# Single-Cell Transcriptomic Analysis Reveals the Molecular Profile of Go-Opsin Photoreceptor Cells in Sea Urchin Larvae

**DOI:** 10.3390/cells12172134

**Published:** 2023-08-23

**Authors:** Maria Cocurullo, Periklis Paganos, Rossella Annunziata, Danila Voronov, Maria Ina Arnone

**Affiliations:** Department of Biology and Evolution of Marine Organisms, Stazione Zoologica Anton Dohrn, Villa Comunale, 80121 Naples, Italy; maria.cocurullo@szn.it (M.C.); periklis.paganos@szn.it (P.P.); rossella.annunziata@szn.it (R.A.); danalvoronov@gmail.com (D.V.)

**Keywords:** opsin, photoreceptors, evolution, sea urchin

## Abstract

The ability to perceive and respond to light stimuli is fundamental not only for spatial vision but also to many other light-mediated interactions with the environment. In animals, light perception is performed by specific cells known as photoreceptors and, at molecular level, by a group of GPCRs known as opsins. Sea urchin larvae possess a group of photoreceptor cells (PRCs) deploying a Go-Opsin (Opsin3.2) which have been shown to share transcription factors and morphology with PRCs of the ciliary type, raising new questions related to how this sea urchin larva PRC is specified and whether it shares a common ancestor with ciliary PRCs or it if evolved independently through convergent evolution. To answer these questions, we combined immunohistochemistry and fluorescent in situ hybridization to investigate how the Opsin3.2 PRCs develop in the sea urchin *Strongylocentrotus purpuratus* larva. Subsequently, we applied single-cell transcriptomics to investigate the molecular signature of the *Sp-Opsin3.2*-expressing cells and show that they deploy an ancient regulatory program responsible for photoreceptors specification. Finally, we also discuss the possible functions of the Opsin3.2-positive cells based on their molecular fingerprint, and we suggest that they are involved in a variety of signaling pathways, including those entailing the thyrotropin-releasing hormone.

## 1. Introduction

The ability to perceive and respond to environmental conditions such as light and temperature variations is fundamental to all organisms. Moreover, many of these conditions undergo cyclic variations, and many organisms, including animals, developed mechanisms with which to anticipate and synchronize their activities to the external periodicities; e.g., circadian and circalunar rhythms [1,2]. One of the main sources of information with which to entrain such rhythms is light [3]. Nonetheless, light is used to drive a variety of other behaviors and physiological processes. Well-known are the roles of light in spatial vision, which require specialized structures, such as eyes or eyespots, and a specific organization of the nervous system [4]; and that responsible for non-directional UV-avoidance behaviors that have been widely described in marine larvae [5].

Light perception is executed by specialized photoreceptor cells (PRCs). In order to increase the photosensitive surface available, PRCs possess different types of membrane extensions [6,7] and can be classified accordingly. Bilaterians mainly possess two types of PRCs: ciliary photoreceptors (c-PRCs) and rhabdomeric photoreceptors (r-PRCs). These two types of photoreceptors are diversified both at morphological and molecular levels, with the latter being characterized by different transcription factor (TF) repertoires guiding their specification and distinct phototransduction cascades [7].

The molecules that allow photoreceptors to translate light stimuli into intracellular signals are proteins called opsins. Animal opsins are membrane proteins, members of the G-protein-coupled receptor (GPCR) family, which are able to detect light stimuli due to a Schiff base linkage to a chromophore [8,9]. The chromophore is usually 11-*cis*-retinal, and the linkage happens at a specific lysine residue situated in the VII helix (K296) in the bovine rhodopsin, the first opsin to be fully characterized at the level of its primary amino acid sequence and 3D structure, and which is often used as reference for comparisons with other opsins [8,10]. The presence of this specific residue, therefore, can be used as a fingerprint indication by which to identify a GPCR as bona fide opsin.

Phylogenetic analyses have revealed a widespread distribution of opsins in metazoans and have classified them into a variety of main groups depending on which phyla were taken into account for the analysis [9,11,12,13,14]. Recent studies have classified opsins into the following groups: ciliary opsins (or c-opsins, containing vertebrate visual pigments), rhabdomeric opsins (or r-opsins, containing for example the melanopsins), neuropsins, peropsins, RGR opsins, Go-opsins, and cnidopsin [13,15,16,17,18,19].

Apart from the opsin classification, the molecular events following the opsin activation have been studied in great detail. When the chromophore absorbs a photon, it undergoes a conformational change, becoming all-*trans*-retinal and activating the protein. The activated opsin subsequently binds a G-protein (guanine nucleotide-binding protein), activating a specific phototransduction cascade depending on which G-protein type is involved; for instance, vertebrate visual opsins recruit Gt proteins with a subsequent decrease in the intracellular cGMP levels, closure of the ionic channels, and final cell membrane hyperpolarization, which eventually inhibits the glutamate neurotransmitter release. In contrast, melanopsins and many invertebrate opsins have been shown to activate a Gq-mediated transduction cascade. This involves a signal cascade initiated by the enzyme phospholipase C (PLC) and eventually induces membrane depolarization (reviewed in [14,20]). Moreover, a phototransduction cascade mediated by Go-opsins was discovered in scallop hyperpolarizing photoreceptors [21]. In contrast, Go-Opsin1 has been reported to mediate depolarization in the annelid *Platynereis dumerilii* [22].

Ciliary and rhabdomeric photoreceptors were understood to be typically used for vision in deuterostome and protostome eyes, respectively. However, the finding of ciliary PRCs in protostomes [23,24,25], rhabdomeric photoreceptors in deuterostomes [26,27,28], and the molecular similarities shared by PRC types among different clades suggests that Urbilateria possesses both photoreceptor types [6,29,30]. Nonetheless, how these two different types of photoreceptors evolved remains an open question. The sea urchin *Strongylocentrotus purpuratus* represents an interesting case through which to address this questions since it belongs to deuterostomes but utilizes both rhabdomeric and ciliary opsins [31,32], and its genome encodes for nine opsins: Sp-Opsin1, Sp-Opsin2, Sp-Opsin3.1, Sp-Opsin3.2, Sp-Opsin4, Sp-Opsin5, Sp-Opsin6, Sp-Opsin7, and Sp-Opsin8 [11]. Despite the description of the wide repertoire of opsins encoded in the sea urchin genome, very little is known about their expression patterns and functions. Recently, the sea urchin extraocular photoreceptor cells in *Paracentrotus lividus* at mature rudiment (i.e., when the adult rudiment formed in the left side of an eight-armed larva is fully mature and ready to go through metamorphosis) and juvenile stages was investigated. In brief, while the mature rudiment expresses only Opsin2-positive and Opsin4-positive r-PRCs, juveniles express all the aforementioned Opsins (Opsin1, Opsin2, Opsin3.1, Opsin3.2, Opsin4, and Opsin5) except Opsin6 and 7 [33]. Moreover, the study of Opsin1 and Opsin4 PRCs in the juveniles of *P. lividus* revealed an expression pattern similar to their orthologous in *S. purpuratus* juveniles [27,34], suggesting conservation of their functions in the two species. Less is known about Opsin expression in sea urchins at larval stages. The Go-Opsin, known as Opsin3.2 in *S. purpuratus* and *P. lividus*, was found to be expressed in two cells located at the sides of the apical organ in *S. purpuratus* [35] and *Hemicentrotus pulcherrimus* [36]. The *Sp-Opsin3.2*-expressing PRCs, which have been previously described as non-directional [35], also express evolutionary conserved developmental transcription factors similar to what is found in the c-PRCs of other animals, typically deploying a c-opsin. However, it remains to be clarified if the regulatory programs of ciliary and Go-Opsin photoreceptor cells derive from the same ancestor photoreceptor cell program and, in this case, how they diversified during evolution.

Sea urchin larvae also express Opsin2 in a few cells localized in the oral and post-oral arms, likely belonging to a mesenchymal cell population [37]. Sea urchin Opsin2 was originally identified as belonging to the echinoderm specific Echinopsin A type [11], but most recently, it has been re-clustered in a new group called Bathyopsins [17]. Nonetheless, nothing else is known about the sea urchin larva Opsin2 photoreceptors in terms of their regulatory state and cell identity.

The single-cell omics approaches have enabled the identification of cell types and helped in reconstructing their evolution at an unprecedented level. Single-cell RNA sequencing (scRNA-seq) has been successfully applied to echinoderm embryonic and larval stages to investigate animal evolution from a cell-type perspective [38,39,40]. In addition, scRNA-seq of *P. lividus* mature rudiments has revealed the presence of distinct PRC populations utilizing an evolutionary conserved photoreceptor regulatory program [33].

Here, we used fluorescent in situ hybridization (FISH) and immunohistochemistry (IHC) to identify and trace, during *S. purpuratus* development, a Go-Opsin (Sp-Opsin3.2)-positive neuronal cell type. Using single-cell transcriptomics, we were able to reconstruct the molecular identity of these photoreceptor cells, contributing to disentangling their evolution and predicting their function. Moreover, we compared the profiles of the *Opsin3.2*-positive cells with the single-cell transcriptomes of the other PRCs present in the *S. purpuratus* larva, the *Opsin2*-positive cells [37]. Through this analysis, we showed that the Opsin3.2 PRCs employ an evolutionary conserved photoreceptor genetic program and are able to produce multiple neurotransmitters, suggestive of multifunctionality; we also partially reconstructed the putative phototransduction cascade activated within these cells, as predicted by single-cell transcriptomics. Furthermore, comparison between the two photoreceptors showed many similarities but also significant differences at the level of TFs expressed and the activated transduction cascade; for example, while the Opsin3.2 PRCs are neurons associated with the apical plate of the larva, thus possibly from ectodermal origin, the Opsin2 PRCs have a mesodermally derived muscle and immune cell identity. Last but not least, we investigated the presence of the circadian rhythm machinery in the Opsin3.2 and Opsin2 PRCs, finding no significant expression of genes that were suggested to be part of the sea urchin master clock [41].

## 2. Materials and Methods

### 2.1. Animal Husbandry

Adult *S. purpuratus* were obtained from Patrick Leahy (Kerckhoff Marine Laboratory, California Institute of Technology, Pasadena, CA, USA) and maintained in circulating seawater at Stazione Zoologica Anton Dohrn in Naples at 15 °C. Gametes were obtained via vigorous shaking the adult sea urchins. The sperm was collected dry using a Pasteur pipette and stored at 4 °C until usage. To collect eggs, females were inverted over a beaker filled with diluted 9:1 (9 parts of Mediterranean Sea seawater and 1 part of distilled water) FMSW. About 20 mL of eggs were fertilized, adding a few drops of sperm diluted 1:10,000. Embryos were transferred in FMSW and reared at 15 °C under a 12 h light/12 h dark cycle. Larval cultures were maintained by exchanging half of the FMSW with fresh FMSW 2 times per week. After 3 days post-fertilization (dpf) pluteus stage, the larvae were fed 3 times per week with the unicellular micro-algae *Dunaliella sp.* at an approximate concentration of 1000 cells/mL.

### 2.2. Fluorescent In Situ Hybridization (FISH) and Immunohistochemistry (IHC)

Whole-mount RNA fluorescent in situ hybridization and combined FISH–IHC were performed as described in [42,43]. Summarizing, specimens at different developmental stages were collected and fixed in Fixative I (4% PFA in 0.1 M MOPS and 0.5 M NaCl) for at least one night at 4 °C. Dubsequently, samples were washed 3 times with MOPS buffer (0.1 M MOPS, 0.5 M NaCl, 0.01% Tween 20, Sigma Corporation, St. Louis, MO, USA)) for 15 min at RT, dehydrated in 70% ethanol, and finally stored at −20 °C until usage. Antisense probes were transcribed from linearized DNA and labeled during transcription using Digoxigenin (Roche, Indianapolis, IN, USA)- or Fluorescein (Roche, Indianapolis, IN, USA)-labeled ribonucleotides following the manufacturer’s instructions. Trh probe, instead, was DNP-labeled as described in detail in [42,43]. Primer sequences used for cDNA isolation and probes synthesis are included in Appendix A. Fluorescent signal was developed via fluorophore conjugated tyramide technology (Perkin Elmer (Waltham, MA, USA), Cat. #NEL752001KT). For combined FISH–HIC, after the tyramide amplification step, samples were incubated in blocking (containing 1 mg/mL bovine serum albumin and 4% sheep serum in PBS) for 1 h at RT, then transferred in primary antibody diluted 1:400 in blocking O/N at 4 °C. Samples were washed 4–6 times with PBS 1×, stained with appropriate Alexa Fluor secondary antibodies (488 rabbit, 555 rat) diluted 1:1000 in blocking, and finally washed 4–6 times with PBS 1×. DAPI (10 mg/mL stock) was added to the samples at a final dilution of 1:10,000 to stain nuclei. Specimens were imaged using a Zeiss (Jena, Germany) LSM 700 confocal microscope, and pictures were analyzed using ImageJ (v1.53v). To stain the Opsin3.2-positive cells, we used an anti Sp-Opsin3.2 gifted by Dr. Robert D Burke. To stain the TRHergic cells, we used a custom antibody produced for us by GenScript, adapting the method used by [44]. Briefly, the Sp-TRH mature amidated peptide (QYPGa) was coupled to an immunogen (Keyhole limpet haemocyanin, KLH) via an N-terminal cysteine. Subsequently the KLH–CQYPGa was used to immunize rabbits. Sera were affinity purified against the antigen by the company, which obtained nine antibody fractions. All the fractions were tested via Elisa with different sea urchin neuropeptides to find the most specific ones, which were then used to perform immunohistochemistry [45].

### 2.3. Larvae Dissociation

Dissociation of the five dpf *Strongylocentrotus purpuratus* plutei into single cells was performed as described in [40]. In brief, larvae were collected, seawater was removed, and larvae were resuspended in Ca^2+^ Mg^2+^-free artificial sea water and then passed to dissociation buffer containing 1 M glycine and 0.02 M EDTA in Ca^2+^ Mg^2+^-free artificial sea water. Larvae were incubated for 10 min on ice and mixed gently via pipette aspiration every 2 min. From that point onwards, the progress of dissociation was monitored. Cell viability was assessed via using propidium iodide and fluorescein diacetate, and only specimens with cell viability ≥90% were further processed. Single cells were counted using a hemocytometer and diluted according to the manufacturer’s protocol (10× Genomics, Pleasanton, CA, USA). Throughout this procedure, samples were kept at 4 °C.

### 2.4. Single-Cell RNA Sequencing and Data Analysis

Single-cell RNA sequencing was performed using the 10× Genomics single-cell capturing system. Specimens from two independent biological replicates were loaded on the 10× Genomics Chromium Controller. Single-cell cDNA libraries were prepared using the Chromium Single-cell 3′ Reagent Kit (Chemistry v3, 10× Genomics, Pleasanton, CA, USA). Libraries were sequenced by GeneCore (EMBL, Heidelberg, Germany) for 75 bp paired-end reads (Illumina NextSeq 500, Illumina, San Diego, CA, USA). ScRNA-seq output reads were aligned and analyzed using Cell Ranger Software Suite 3.0.2 (10× Genomics). The genomic index was made in Cell Ranger using the *S. purpuratus* genome version 3.1 [46,47]. Cell Ranger output matrices for two biological replicates were used for further analysis in the Seurat v4 R package [48]. The analysis was performed according to the Seurat scRNA-seq R package documentation [48,49]. Briefly, variable genes were identified using the VST method. Finally, nearest neighbor (SNN) graph and uniform manifold approximation and projection (UMAP) were computed to identify the clusters. For this analysis, data from 3 dpf scRNA-seq [40,50] and 5 dpf datasets were merged with the method described in [51]. Subsequently, the clusters containing *Sp-Opsin3.2* and *Sp-Opsin2* were selected. A detailed description of the methods can be found in the Appendix A.

## 3. Results

### 3.1. Expression Pattern of Opsin3.2 (Go-Opsin) during Larval Development

To investigate the expression pattern of the *Sp-Opsin3.2* gene during *S. purpuratus* larva development, we combined IHC and FISH performed on larvae collected at different developmental stages, from the early two-armed pluteus (3 dpf) to the eight-armed pluteus (5 wpf). At the 3 dpf pluteus stage, *Sp-Opsin3.2* FISH revealed the presence of one or two positive cells located in the area above the mouth, distally to the larval apical organ (Figure 1A–B′). The apical organ is a sensory organ present in many marine larvae involved in settlement and metamorphosis [52,53,54]. In sea urchin larvae, it is located between the oral arms, just above the mouth, and mainly consists of a group of serotonergic neurons [31,52,55]. In the 4 dpf larvae, the number of cells positive for *Sp-Opsin3.2* transcripts increases to two or three (Figure 1C–D′), while at 5 dpf, they reach a total number of three to four cells (Figure 1E,F). At the six-arm pluteus stage (4 wpf), we could detect the presence of Opsin3.2 ganglia (Figure 1G,G′), situated in the same region with respect to the apical organ right above the mouth and below the joint with the oral arms. These ganglia remain up to the eight-arm competent to metamorphosis pluteus larva (5 weeks post fertilization) (Figure 1G–H″).

Previous studies have demonstrated the presence of neurons that produce the thyrotropin-releasing hormone neuropeptide (TRH) in cells located in similar regions as the Opsin3.2-positive neurons [40,56]. In order to understand whether the TRH-positive cells (TRHergic neurons) are in close proximity to the Opsin3.2 PRCs, and if TRH could be produced by the Opsin3.2 PRCs, we performed double FISH and double IHC for Sp-Opsin3.2 and Sp-Trh. Interestingly, the co-localization of both proteins and *Sp-Trh* transcripts in the same neurons of the 3 dpf pluteus larva (Figure 1B,B′) indicate that Opsin3.2 PRCs are able to produce the neuropeptide TRH. Similarly, at 4 and 5 dpf pluteus stages, all the Sp-Opsin3.2-positive cells are also TRHergic, as shown by *Sp-Opsin3.2/Sp-Trh* double FISH combined with Sp-TRH immunostaining (Figure 1F) and TRH/Opsin3.2 double immunostaining combined with *Sp-Trh* FISH (Figure 1B–D′)

### 3.2. Characterization of the Go-Opsin3.2- and the Opsin2-positive Cell Types at a Single Cell Resolution

ScRNA-seq is a powerful technique, which allows us to identify cell clusters or populations, finely disentangle their regulatory state (i.e., TFs expressed), and predict possible functions and interactions among cell types (based on which genes, signaling molecules, or receptors are expressed). Such an approach was successfully applied to identify and characterize sea urchin larval cell populations [40,50,57] at the early stages. Compared to the 3 dpf pluteus, more Opsin3.2-positive cells are present in the 5 dpf pluteus (See Figure 1). This observation prompted us to perform scRNA-seq on *S. purpuratus* 5 dpf pluteus larvae and to combine it with the already available 3dpf pluteus dataset to investigate PRC diversity and the Go-Opsin3.2 molecular fingerprint. Our analysis included single-cell transcriptomes from 32,116 cells and resulted in the identification of 21 clusters corresponding to distinct cell types or closely related cell type families (Figure 2B,C). The quality of the clustering analysis was assessed using gene markers previously shown to label distinct cell type families at the 3 dpf *S. purpuratus* pluteus stage: *Sp-Fbsl_2* and *Sp-Btub2/3* (ciliary band); *Sp-Hbn* and *Sp-Frizz5/8* (apical plate); *Sp-Spec2a* and *SPU_006199* (aboral ectoderm); *Sp-SynB* and *Sp-Chrna9_4* (neurons); *Sp-Mhc* and *Sp-Mlckb* (esophageal muscles), *Sp-Nan2* and *Sp-Vasa* (coelomic pouches); *Sp-MacpfA2* and *Sp-Pks1* (immune cells); *Sp-185/333B3d* and *Sp-Fcoll/II/IIIf* (blastocoelar cells); *Sp-Msp130* and *Sp-Sm37* (skeletogenic cells); *Sp-Hox11/13b*, *Sp-Cdx* and *Sp-Rfxc1l* (posterior gut); *Sp-Nkx6.1* and *Sp-Pdx1* (pyloric sphincter); *Sp-Hnf4*, *Sp-ManrC1a* (stomach); *Sp-Ptf1a* and *Sp-Cpa2L* (exocrine pancreas-like); *Sp-Ahrl* and *Sp-Trop1* (cardiac sphincter); and *Sp-Alpi* and *Sp-Brn1/2/4* (esophagus/oral ectoderm). Plotting the average expression of the aforementioned gene markers resulted in a meaningful clustering, reconstructing most of the well-known larval cell types (Figure 2B,C).

To identify which cell clusters express our Go-Opsin of interest, the average expression of *Sp-Opsin3.2* was plotted in each cluster (Figure 3A,B). These data show that only the Neurons (4) cluster contains cells that highly express the Opsin3.2 (Figure 3B). Notably, this is also the cluster with the most significant *Sp-Trh* expression levels (Figure 3A). We next plotted the *Sp-Opsin2* average expression and found it enriched in esophageal muscles and immune cells, suggesting a mesodermal origin for these cells, as already shown in another sea urchin species [37]. Moreover, our single-cell data indicate that the *Sp-Opsin2*-positive cells belonging to the immune cells cluster could possibly be pigment cells, as judged via the co-expression of the pigment cell marker *Sp-Pks1* in (Figure 3A). For our analysis, we choose to focus on the Opsin2 PRCs belonging to the immune ells population (Figure 3B); therefore, all of the following plots including *Sp-Opsin2*-positive cells have been produced by selecting only the *Pks1/Opsin2*-positive cells.

Finally, to confirm the quality of our data, we plotted the average expression of transcription factors that were found through FISH to be either co-expressed with *Sp-Opsin3.2* cells or in the *Sp-Opsin3.2* region [18]. The list includes, but is not limited to, *Sp-Otx*, *Sp-Tbx2/3*, and *Sp-Six3*, which have been suggested to be part of an ancestral photoreceptor specification code (Figure 3C) [6]. The average expression of the selected genes was plotted in all the cell clusters, and the main accumulation was found in the apical plate and Neuron (4) cells, while the immune cells had high expression only of the transcription factor *Sp-Id* (IDs are a family of helix–loop–helix proteins involved in regulating cell proliferation and differentiation) and low expression levels of *Sp-Tbx2/3*.

### 3.3. Molecular Signature of Photoreceptor Cells in S. purpuratus Larvae

To gain additional information on how the *Sp-Opsin3.2*-positive cells work, we exploited the scRNA-seq data to explore their molecular signature and compared it to the molecular signature of the Opsin2 PRCs. In particular, the analysis was specifically targeted to the *Sp-Opsin3.2*-expressing cells and the *Sp-Opsin2*-positive cells belonging to the immune cell cluster.

Go-Opsin photoreceptors in invertebrates have been found to be either depolarizing, as in the worm *P. dumerilii* [22], or hyperpolarizing, as in scallops [21]. To determine which type of signaling is activated by the stimulation of the Opsin3.2 PRCs in the sea urchin larva, the expression levels of all the *S. purpuratus* genes that are known to be part of phototransduction cascades in other organisms were plotted in the *Sp-Opsin3.2*-positive cells (Figure 4A). Such gene expression profiles were plotted also in the *Sp-Opsin2*-positive cells belonging to the immune cells to investigate the differences between the two cascades (Figure 4A). Unexpectedly, the G-protein types most highly expressed in the Go-Opsin PRCs are the *Sp-Gs*, *Sp-Gi* and *Sp*-*GbA*, while the G proteins annotated as *Sp-Go* and *Sp-GoI* (https://www.echinobase.org/entry/ (accessed on 1 September 2022)) are lowly expressed, suggesting that the first types of G proteins activated by the Go-Opsin3.2 are Gs, Gi, and GbA. Nonetheless, other putative components of the phototransduction cascade are only lowly expressed, and it is therefore difficult to make any further prediction on the activated cascade. For example, many genes annotated as phosphodiesterases (PDEs) are expressed in a small percentage of cells and at low levels. The expression of PDEs might be indicative of a hyperpolarizing cascade, similarly to what happens in mammalian eye photoreceptor cells (see the example in [14]). Looking at the genes from the phototransduction cascade expressed in the *Sp-Opsin2*-positive cells, the overall picture is relatively similar to the Opsin3.2 PRCs in terms of G-protein expressed and their expression levels; however, the *Sp-Opsin2*-positive cells do not show any significant expression of the other putative genes involved in the phototransduction cascade.

Furthermore, we investigated the possibility that the Opsin3.2 protein might be involved in the entrainment of circadian clock (Figure 4B). The *S. purpuratus* genome encodes for several putative orthologues of the components of the canonical animal circadian clocks; e.g., clock (*Sp-Clock*), timeless (*Sp-Tim*), brain and muscle Arnt-like protein (*Sp-Bmal*), cryptochromes (*Sp-Dcry* and *Sp-Vcry*), and hepatic leukemia factor (*Sp-Hlf*) [31]. Interestingly, of the canonical core genes, the Opsin3.2 PRCs express only *Sp-Clock* and at low levels (Figure 4B). Nonetheless, they also express *Sp-Shaggy*, which, in other species, is responsible for the phosphorylation of Tim and therefore regulates its function. The Opsin3.2 PRCs also highly express *Sp-Hlf,* which was found to oscillate during the diel cycle (but not in a free-running condition) [41]. Finally, *Sp-Sim*, *Sp-Slimb*, *Sp-Hey*, and *Sp-Nfil3* are expressed at low levels in the Opsin3.2 PRCs. On the contrary, the Opsin2 PRCs (Figure 4B) express only *Sp-Hlf* and *Sp-Shaggy* and at levels comparable to those found in the *Sp-Opsin3.2*-positive cells. In addition, they have low levels of *Sp-6_4photolyase* (photolyases are enzymes that repair DNA damages caused by UV light, [58]) and *Sp-Ncoa*.

Based on the evidence that the *Sp-Opsin3.2*-positive cells are neurons belonging to the Neuron (4) cluster, we investigated which type of neuronal genes are found in these PRCs, taking advantage of the available comprehensive characterization of the larval nervous system [40] (Figure 4C). Notably, the *Sp-Opsin3.2* cells express not only genes involved in photoreceptor specification (*Sp-Otx*, *Sp-Six3*, and *Sp-Rx*) but also genes involved in neuron/anterior neuroectoderm patterning (*Sp-Nkx2.1*, *Sp-Hbn*, *Sp-Soxb1*, *Sp-SoxC,* and *Sp-Brn1/2/4*). Looking at the signaling molecules, the gene with the highest expression level is the *Sp-Trh*, neuropeptide, which is evidence for the double function of the Go-Opsin PRCs as both photoreceptor and neurosecretory cells. Moreover, it has been shown, both through combining IHC and FISH and by scRNA-seq, that the TRHergic cells also produce another neuropeptide, Sp-Salmfap [40,56].

To conclude, we found also that both *Sp-Opsin3.2*- and *Sp-Opsin2*-positive cell populations express the TFs *Sp-Otx* and *Sp-Id*, which are fundamental for general photoreceptor specification; while *Sp-Rx*, expressed in ciliary photoreceptors in many animals [18], is expressed only by the *Sp-Opsin3.2*-positive cells (Figure 4E).

## 4. Discussion

The main aim of this work was to characterize the Opsin3.2 photoreceptors found in *S. purpuratus* larvae. Nonetheless, during our analysis, we encountered a second photoreceptor cell type known at this stage, i.e., the one expressing the *Sp-Opsin2*. Opsin2 PRCs have been already described, not only in *S. purpuratus* [18] but also in other sea urchin species (namely, *Hemicentrotus pulcherrimus* [37]), and have been shown to be a peculiar photoreceptor group, probably rising from a mesenchymal cell population. Indeed, our data also found a group of *Sp-Opsin2*-positive cells belonging to the Immune cell cluster. Comparison of the molecular signature of Opsin3.2 and Opsin2 PRCs showed many similarities in the TFs involved in their specification (both express *Sp-Soxb1*, *Sp-Soxb2*, *Sp-Id*, *Sp-Otx*, and *Sp-Hbn*, and overall, they share more than 80% of expressed TFs, Figure 4D) in the activated phototransduction cascade and in genes deployed for circadian rhythm establishment, with minor differences. Subsequently, despite the evidence that Opsin2 PRCs do not belong to a neuronal population, they still express typical neuronal markers. Major differences arose when we looked deeply into the molecular identity of the two PRCs in terms of the neuropeptides, receptors, and other TFs involved in neuronal specification; for example, Opsin2 PRCs do not express the *Sp-Trh* neuropeptide and the *Sp-Drd1* dopamine receptor. Nonetheless, further comparison of the differences between the two photoreceptor cell types was outside the scope of this work, and we will not further discuss the collected data concerning *Sp-Opsin2*-positive cells.

### 4.1. Opsin3.2 PRCs Characterization during S. purpuratus Larval Development

The presence of two Opsin3.2-positive cells at the sides of the apical organ in the *S. purpuratus* sea urchin larva had already been reported [18,35], but no information was available about their development from early- to late-larval stages. Therefore, first, we combined immunohistochemistry and fluorescent in situ hybridization to identify the Opsin3.2-producing cells in larvae ranging from early four-armed to late eight-armed pluteus stages (Figure 1). Interestingly, the number of Opsin3.2 cells appeared to increase during larval development in a non-stereotypical way, going from one–two cells at early pluteus to two clusters of up to six cells bilaterally distributed at either side of the apical organ at the late pluteus stage (4–5 wpf) (Figure 1). Moreover, combined staining of *Sp-Opsin3.2* and *Sp-Trh* provided the first evidence for a dual sensory/neurosecretory role for those cells. These cells also display a bipolar morphology, with a cilium-like structure hosting the Go-Opsin molecules on one side and the axon transporting the TRH neuropeptide on the other side of the cell. To prove that these cells are ciliary PRCs, the morphology of the ciliated structure requires further investigation via electron microscopy. Nonetheless, our analysis, despite covering the majority of the sea urchin developmental stages, did not cover early embryonic stages. This is because no *Sp-Opsin3.2* or *Sp-Opsin2* expression was found before 3 dpf via IHC, FISH, or scRNA-seq. Moreover, we did not explore later stages of development, e.g., metamorphic plutei, juveniles, and adults, because this was beyond the scope of this work.

### 4.2. Opsin3.2 PRCs Utilize an Ancestral Gene Regulatory Toolkit

Furthermore, we investigated the molecular fingerprint of the Sp-Opsin3.2 photoreceptors and showed that they utilize an ancestral regulatory toolkit; that is, they express TFs required for neuron (*Sp-SoxC* and *Sp-Brn1/2/4*) and anterior neuroectoderm (*Sp-Hbn*, *Sp-Six3*) specification in sea urchins [59]. Moreover, the expression of *Sp-Otx* and *Sp-Six3* supports the ancestral module suggested by [6] to be present in the precursor of all animal photoreceptors. According to this hypothesis, the ancestral photoreceptor cells utilized a variety of opsin types (at least nine in the Bilaterian ancestor [17]) and gave rise to the two sister cell types known as ciliary and rhabdomeric photoreceptors. Each of them also co-opted specific TFs such as Rx in ciliary photoreceptors. Intriguingly, Sp-Go-Opsin3.2 PRCs also express the TF *Sp-Rx* and have been described as belonging to the ciliary type [18]. The hypothesis for the common evolutionary origin of ciliary and rhabdomeric photoreceptors is supported by the evidence that *P. dumerilii* possesses rhabdomeric photoreceptors expressing not only r-opsins but also Go-Opsin1 both in the adult eye and in the larval eyespot [22]. This latter opsin type is most commonly associated with ciliary-type PRCs (e.g., in scallops and sea urchins [18,21]. This hypothesis is also supported by the existence of a r-PRCs utilizing both xenopsin and r-opsin in the larval eye of the mollusk *Leptochiton asellus* [19]. However, whether Urbilateria, the common ancestor of Bilateria, already possessed both ciliary and rhabdomeric photoreceptors, or if it still had a single bimodal ciliary/rhabdomeric precursor cell, is not yet clear. Only analyzing the regulatory toolkit, the morphology, and the opsin repertoire of photoreceptor cells in diverse early branching bilaterians animals will help disentangle this mystery.

### 4.3. Opsin3.2 PRCs as Sensory and Neurosecretory Unit

In addition to their photoreceptor signature, Opsin3.2 PRCs are also able to produce a variety of signaling molecules, mostly neuropeptides (Sp-TRH, Sp-An, Sp-FSALFa/Salmfap, and Sp-Nesfatin) and acetylcholine [40,56]. Therefore, they look to be both sensory and neurosecretory cells.

It has been proposed that the Urbilateria already possessed a simple nervous system containing cells which were both sensory and neurosecretory. In animals with a more complex nervous system, the sensory signal collected by a specified sensory cell is transferred to the central nervous system through a series of interneurons. Here, the information is integrated with the signals received by other sensory cells, and the final output is then transferred to an effector neuron which might be a motor neuron or an endocrine cell, for example. In less complex nervous systems, such as the case of Urbilateria, on the contrary, one cell can both collect the sensory stimulus and elicit the response, thus producing a very simple circuit or a minimal module capable of controlling behavioral or physiological process [60,61]. Among these minimal modules, photosensitive neuroendocrine cells could control neurohormone secretion under particular light conditions [61]. The Opsin3.2 cells could be the descendants of such cells, retained in the sea urchin larva. Intriguingly, a cluster of non-visual photoreceptors expressing the vertebrate ancient long opsin b (valopb) was found in the brain of adult *Danio rerio* [62], suggesting an interesting evolutionary scenario for an ancestral TRHergic/photoreceptor cell cluster, which could be tested by looking for the existence of TRHergic/Opsin-positive neurons in other phyla.

### 4.4. Opsin3.2 PRCs Possible Function(s)

Considering the sensory–neurosecretory nature of the Opsin3.2-positive cells, it is even more interesting to understand their function. Since these are photoreceptors, the first hypothesis is that they might be involved in controlling swimming behavior in response to light. In previous works, it has been suggested that light response activated by the Go-Opsin3.2 PRCs might be directional, i.e., that the larva is able to distinguish from which direction light hits the photoreceptor [18]. However, in all known cases of 3D phototaxis in the water column, the cells containing shading pigments are in close juxtaposition to the photosensitive cells (as reviewed by [63]). Sea urchin pigment cells are not fixed in their position; they migrate in response to infections [64,65], thus representing an unreliable source of shading.

Nonetheless, non-directional perception of light intensity can be used by aquatic animals to adjust their vertical position in the water column [22] and to avoid high UV levels [5,29,66]. Interestingly, Yaguchi and colleagues [37] showed that *Hemicentrotus pulcherrimus* and *Temnopleurus reevesii* larvae change their swimming behavior in response to strong photoirradiation. Although the authors seem to exclude an involvement for the Go-Opsin in this response, thorough experiments including knock-down or knock-out of this opsin exposed to different wavelengths are necessary in order to assess the role of these PRCs in light-mediated swimming behaviors. Moreover, in a previous paper, Yaguchi and collaborators [36] provide evidence for Go-Opsin PRCs controlling sphincter opening in response to light through a serotonergic signaling in *H. pulcherrimus* larvae, and it would be interesting to test if this is also a conserved function also *S. purpuratus* larvae.

Intriguingly, the Opsin3.2-positive cells in our data are also predicted to express the dopamine receptor *Sp-Drd1*. Since dopamine signaling is involved in mediating phenotypic (developmental) plasticity in response to food availability [67], we can hypothesize that the Sp-Opsin3.2 PRCs could control this process. Another possibility is that the light stimulus perceived by the Go-Opsin3.2 is used to entrain a circadian clock. To investigate this possibility, we looked at the genes putatively involved in this process (Figure 3D). Our scRNA-seq data do not show a prominent expression of putative core genes in the Sp-Opsin3.2 PRCs in the 3 and 5 dpf *S. purpuratus* larva, thus suggesting that the Opsin3.2 cells are involved in light perception more so than in circadian rhythm regulation.

While all the evidence collected so far contributes to building hypotheses on the function of the Opsin3.2/TRHergic cells, functional experiments to knock-down or knock-out Opsin3.2, coupled with phenotypic and behavioral observations, are necessary in order to comprehend the function of the Opsin3.2/TRHergic cells in sea urchin larvae.

## 5. Conclusions

The sea urchin *S. purpuratus* larva feature a set of bilaterally symmetrical photoreceptor cells expressing the Go-Opsin3.2. This cell type deploys an ancient conserved regulatory module for photoreceptors specification. This makes them an important component in the reconstruction of photoreceptor/opsin system evolution, especially if combined with the analysis of photoreceptors in early branching animals. Additionally, the wide set of neuropeptides, signaling molecules, and receptors expressed by these cells (including for example, TRH, RX, Salmfap, An, and Drd1) (Figure 5) strongly suggests their involvement in multiple processes. Nonetheless, knock-down and/or knock-out experiments are required in order to really understand the process(es) in which the Opsin3.2 PRCs are involved.

## Figures and Tables

**Figure 1 cells-12-02134-f001:**
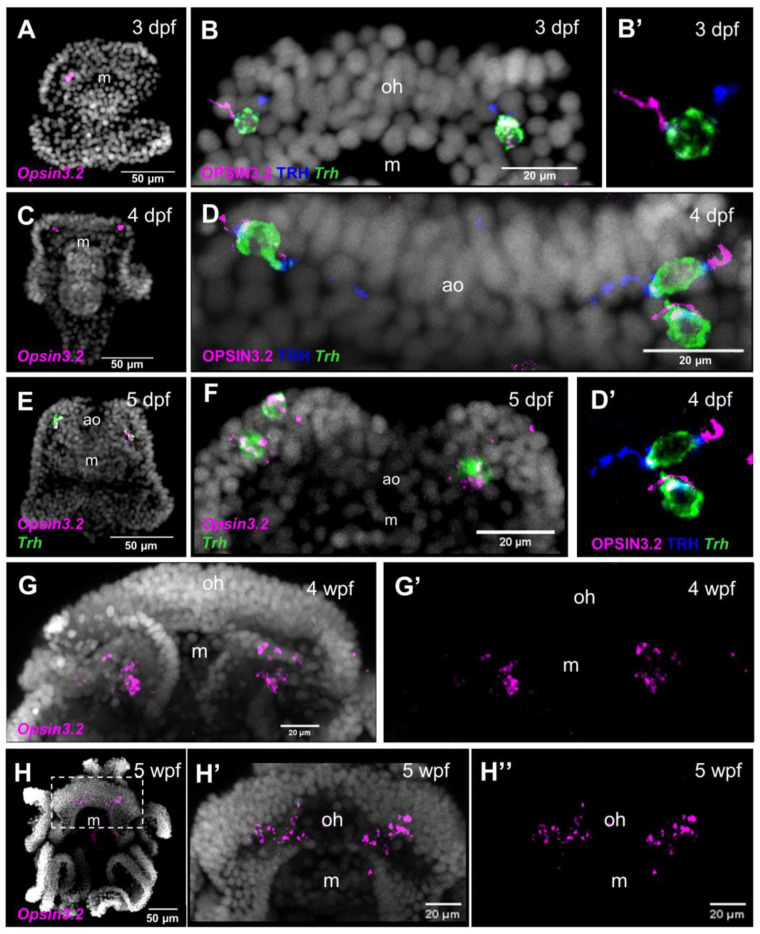
Molecular characterization of the Opsin3.2 cells at different developmental stages. (**A**) FISH of *Sp-Opsin3.2* stains 1 cell at 3 dpf. (**B**) FISH for *Sp-Trh* was combined with double TRH/Opsin3.2 immunolocalization at 3 dpf, highlighting the bipolar structure of these sensory/neurosecretory neurons, with the Opsin3.2 localized on a ciliated-like structure located on the external side of the larva. The TRH peptide, instead, is concentrated in the opposite side of the cell, and it appears to be transported along the projections of the cells (directed toward the larva apical organ). (**B′**) Details of the cell showed in (**B**) on the left. (**C**) FISH of *Sp-Opsin3.2* at 4 dpf. (**D**,**D′**) FISH for *Sp-Trh* was combined with double TRH/Opsin3.2 immunolocalization at 4 dpf. All three cells have a bipolar organization. (**E**,**F**) Double FISH of *Sp-Opsin3.2* and *Sp-Trh* at 5 dpf stains three–four cells. FISH of *Sp-Opsin3.2* at 4 wpf (**G**,**G′**) and 5 wpf (**H**–**H″**) detect two clusters of cells located at the base of the oral arms. All images are full projections of merged confocal Z sections. Nuclei are shown in white. Abbreviations: ao, apical organ; m, mouth; oh, oral hood.

**Figure 2 cells-12-02134-f002:**
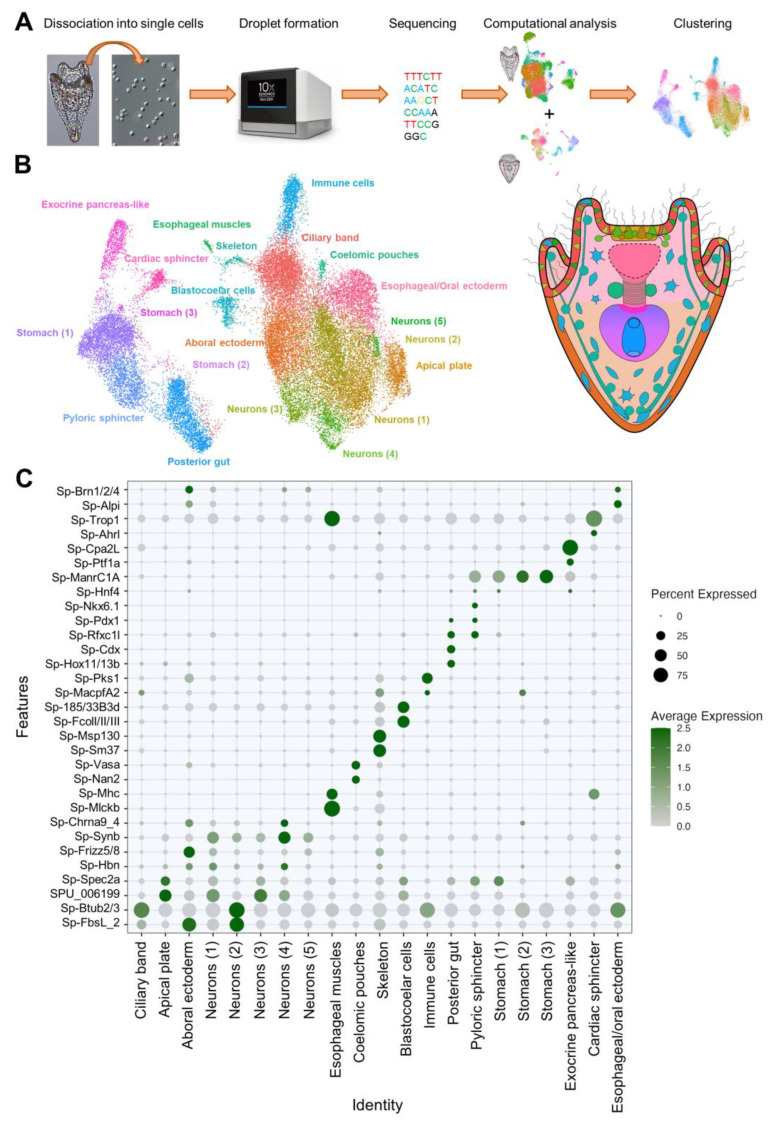
Cell type atlas of the *S. purpuratus* larvae at 3 and 5 dpf pluteus stages. (**A**) Schematic representation of the scRNA-seq pipeline. (**B**) UMAP representing the cell clustering obtained through harmony of the 3 and 5 dpf pluteus single-cell datasets and their localization in a schematic representation of a four-armed *S. purpuratus* larva. (**C**) Dotplot showing the average expression of a subset of genes used as markers to annotate the different cell clusters.

**Figure 3 cells-12-02134-f003:**
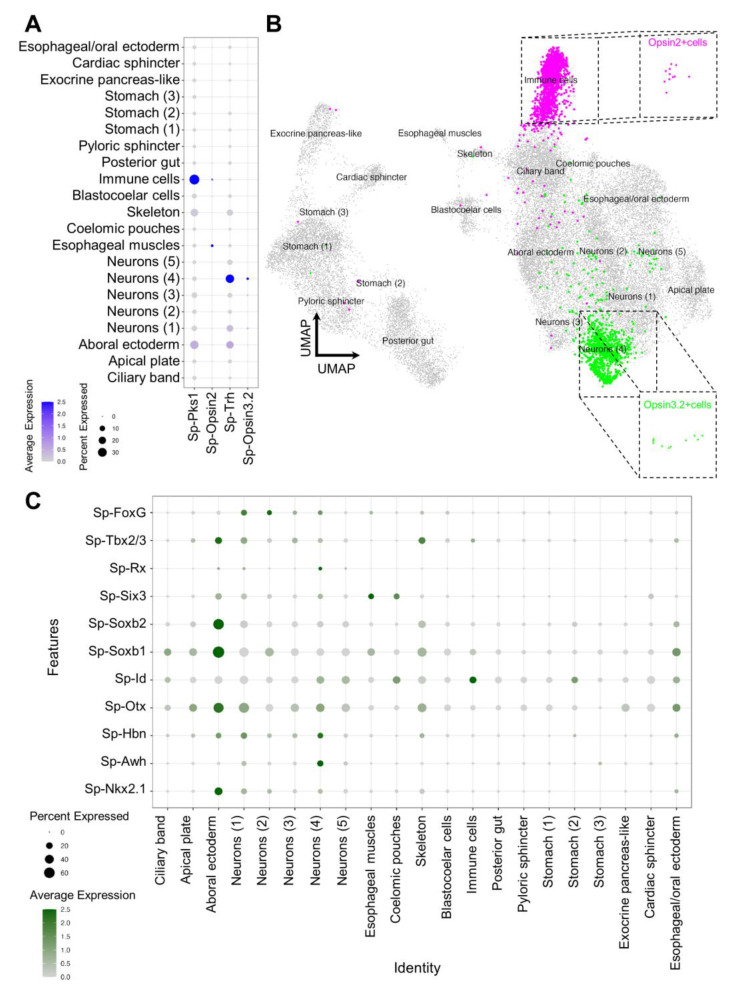
Identification of the Opsin3.2 and Opsin2 PRCs. (**A**) Dotplot showing the expression of *Sp-Opsin3.2* and *Sp-Opsin2* and of the neuropeptide *Sp-Trh* and of the immune cell marker *Sp-Pks1*. (**B**) UMAP illustrating that Opsin3.2 PRCs have been isolated from the Neuron (4) cluster, while Opsin2 have been selected from the immune cells cluster. (**C**) Dotplot showing the expression pattern of genes selected from [18].

**Figure 4 cells-12-02134-f004:**
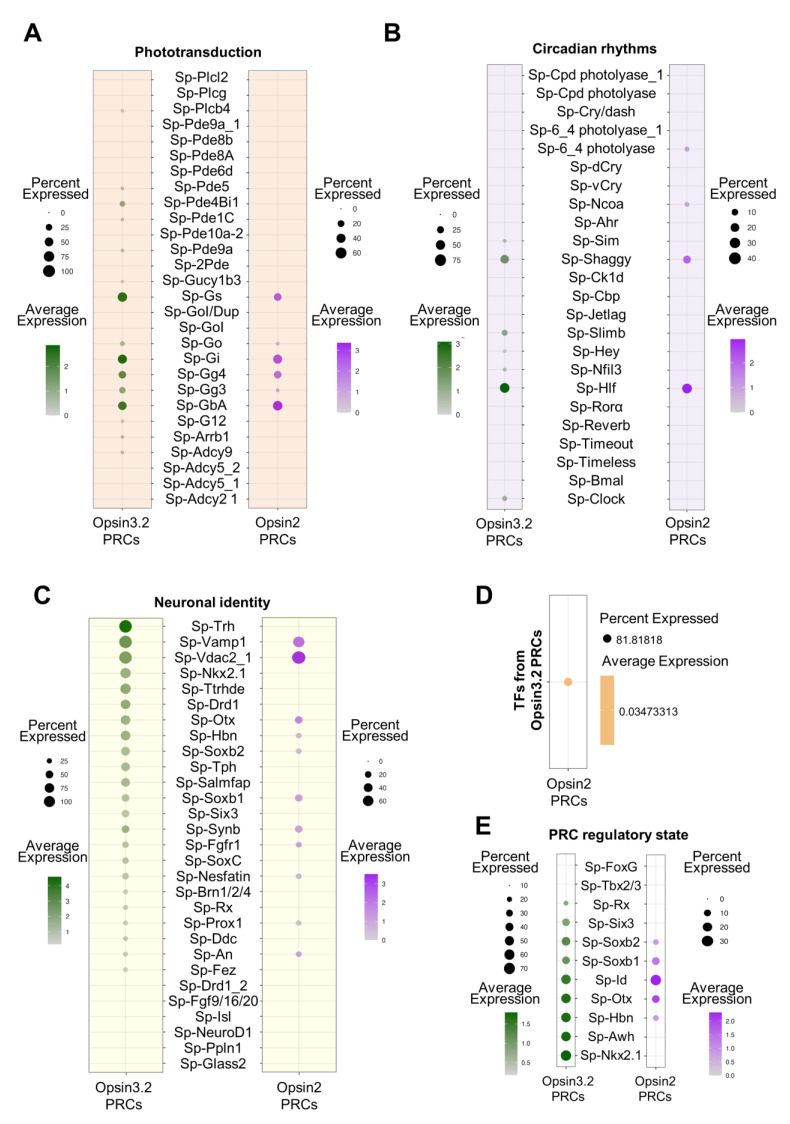
Molecular signature of *S. purpuratus* PRCs at pluteus stages. (**A**) Dotplot showing the average expression of genes putatively involved in the phototransuction cascade activated by the opsins stimulation. (**B**) Dotplot showing the average expression of genes involved in the regulation circadian rhythms. (**C**) Dotplot showing the average expression of genes selected to further describe the molecular identity of the Opsin3.2 and Opsin2 PRCs. (**D**) Dotplot showing the percentage of TFs common to the two PRCs. (**E**) Dotplot showing the average expression of known TFs involved photoreceptor specification.

**Figure 5 cells-12-02134-f005:**
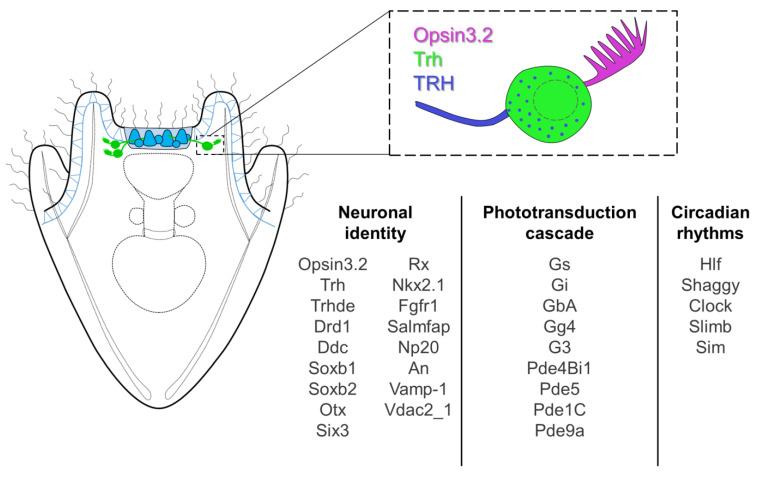
Opsin3.2 PRCs summary. Schematic reconstruction of the morphology, topology and molecular signature of the Opsin3.2 PRCs.

## Data Availability

The raw reads for the sequencing data obtained for this study and the RDS file are available at NCBI Gene Expression Omnibus (accession number GSE240882).

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
