# Peer review of "Single-Cell Transcriptomic Analysis Reveals the Molecular Profile of Go-Opsin Photoreceptor Cells in Sea Urchin Larvae"

_cells, 2023, doi:10.3390/cells12172134_

Round 1

Reviewer 1 Report

In this manuscript, the authors focused on Opsin3.2 in sea urchin larvae, and investigated how Opsin3.2 photoreceptor cells develop by some using immuno-detection and Single-cell transcriptomics. Through these analyses, the authors found that Opsin3.2 PRCs are involved in several signaling pathways including thyrotropin-hormone releasing. In addition, they also focused on another PRC, Opsin2-cell, and analyzed its transcriptomics, compared with those of Opsin3.2 PRC. Based on these results, the authors discussed the evolution of ciliary PRCs in sea urchins. Overall, analyses and interpretations were of interest to even non-sea urchin scientists and the text is well written. I basically recommend this published in this journal, but a few concerns need to be fixed before that. 

Images of Figure 1 look strange to readers. They look like that all confocal planes are stacked for nuclei but only a few surfaces are for PRCs. Is this normal way in handling microscope images in the field? Readers need to know whether the signals are specific or not. 

The authors use Hbn and Six3 for anterior neuroectoderm makers, but is this OK for 3 or older days larvae? 

In line 220, I can see a red dot in the text. 

In line 316, what is P. dumerilii? 

In line 361, Opsin3 should be Opsin2. 

In lines 393-397, these sentences make readers expect that the authors revealed the origin of Opsin3.2 PRCs but actually not. 

In line 444, reference style? 

Author Response

We would like to thank the reviewer for their comments. Point-by-point response to Reviewer 1's comments follows:

Images of Figure 1 look strange to readers. They look like that all confocal planes are stacked for nuclei but only a few surfaces are for PRCs. Is this normal way in handling microscope images in the field? Readers need to know whether the signals are specific or not. 

We are sorry if this was the impression given, but all images shown in Figure 1 are full projections of every channel. This is evident from  the background dots in C around the gut area, in the lower center of D (just above the 5 dpf). Also in G’, H, and H’’. Probably the wrong impression is given by the fact that the nuclei are many, with strong signals, while the opsin/TRH expressing cells are only a few, and the signal is very specific.

The authors use Hbn and Six3 for anterior neuroectoderm makers, but is this OK for 3 or older days larvae? 

Six3 and Hbn are markers for the anterior neuroectoderm in S. purpuratus larvae at 3 dpf according to ref [60] Angerer, L.M.; Yaguchi, S.; Angerer, R.C.; Burke, R.D. The Evolution of Nervous System Patterning: Insights from Sea Urchin Development. Development 2011, 138, 3613–3623, doi:10.1242/dev.058172.

For older larvae, we found this to be still valid, see Paganos, Periklis. «Cell Type Diversity During Sea Urchin Development: A Single-Cell Approach to Reveal Different Neuronal Types and Their Evolution». The Open University, 2021.

In line 220, I can see a red dot in the text

We thank the reviewer for the warning, we fixed this issue.

In line 316, what is P. dumerilii? 

We now included the extended version of the name of the annelid Platynereis dumerilii in the introduction (line 76).

In line 361, Opsin3 should be Opsin2. 

We thank the reviewer and fixed the issue. 

In lines 393-397, these sentences make readers expect that the authors revealed the origin of Opsin3.2 PRCs but actually not. 

We agree with the reviewer and rephrased the first sentence to: “The presence of two Opsin3.2 positive cells at the sides of the apical organ in the S. purpuratus sea urchin larva was already reported [18,35] but no information was available about their development from early to late larval stages.”

In line 444, reference style? 

We fixed this issue with the style of the reference which is now reference [62].

Reviewer 2 Report

The authors apply single-cell RNA sequencing to elucidate the molecular properties of photoreceptor cells in sea urchin larvae. This provides new insights into an understudied cell type, incrementally advancing knowledge of photoreceptor diversity. However, the focus on investigating photoreceptor evolution is not highly original, and the findings represent a limited advance for the broader field, being most relevant to researchers specifically interested in this area. 

The manuscript is clearly written and technically sound overall. I have no major critiques of the methodology or analyses. My main suggestions for the authors involve:

- Enhancing the clarity and legibility of figures. 

- Using standard gene/protein nomenclature consistently throughout.

- Proofreading to fix minor typos like "Sp-Osin3.2."

- Discussing any limitations or caveats regarding developmental stage examined.

To enhance the clarity and legibility of figures, I recommend:

- Increasing font sizes for axis labels, legends, and text in charts/plots to at least 10-12 pt so they can be easily read without zooming.

- In Figure 3 specifically, some characters appear masked by adjacent panels. Adjusting the layout could help improve readability.

- Verifying all text and features in illustrations is sufficiently sharp and large.

Additionally, the title could be simplified to more precisely convey the key techniques, cell type, and findings. For example: 

"Single-cell transcriptomic analysis reveals the molecular profile of Go-Opsin photoreceptor cells in sea urchin larvae."

Moreover, suggest that the methods and peripheral results that could potentially go in supplementary info include:

- The detailed statistical tests for the scRNA-seq analysis (keep summary in main text).

- The circadian rhythm gene analysis, since the link to Opsin3.2 cell function is unclear.

Finally, some long complex sentences, like this example in the Discussion : "In any case, intriguing is the expression, although in a small percentage of cells and at low levels, of many genes annotated as phosphodiesterases (PDEs), which might be indicative of a hyperpolarizing cascade, similarly to what happens in mammalian eye photoreceptor cells..." could be divided into shorter sentences to enhance readability. 

In summary, this paper solidly demonstrates the molecular signature of an understudied photoreceptor cell type, but the research represents an incremental advance in a specialized area rather than a major leap forward. With minor revisions to highlight the most salient results, it will make a nice addition to the literature on photoreceptor diversity.

While no significant issues were noted, I recommend the authors proofread the manuscript carefully to fix any minor errors and ensure consistency. Also, having a co-author help copyedit the text could catch small mistakes.

Author Response

We would like to thank the reviewer for their comments. Point-by-point response to Reviewer 2’s comments follows:

- Enhancing the clarity and legibility of figures. 

We increased the font size in all graphs, whenever it was possible without compromising the quality of the graphs’ organization.

- Using standard gene/protein nomenclature consistently throughout.

We thank the reviewer for this comment and proofread the manuscript to fix this issue.

- Proofreading to fix minor typos like "Sp-Osin3.2."

We thank the reviewer for this comment and proofread the manuscript to fix these typos.

- Discussing any limitations or caveats regarding developmental stage examined.

To answer this point, we included a few more sentences at the end of chapter 4.1. 

Nonetheless, our analysis, despite covering the majority of the sea urchin developmental stages, did not cover early embryonic stages. This is because no Sp-Opsin3.2 or Sp-Opsin2 expression was found before 3 dpf by IHC, FISH or scRNA-seq. Moreover, we didn’t explore later stages of development, e.g. metamorphic plutei, juveniles and adults, because this was beyond the scope of this work.”

To enhance the clarity and legibility of figures, I recommend:

- Increasing font sizes for axis labels, legends, and text in charts/plots to at least 10-12 pt so they can be easily read without zooming.

We increased the font size in all graphs, whenever it was possible without compromising the quality of the graphs’ organization

- In Figure 3 specifically, some characters appear masked by adjacent panels. Adjusting the layout could help improve readability.

We thank the reviewer for noticing this issue and fixed the mistake.

- Verifying all text and features in illustrations is sufficiently sharp and large.

Done

Additionally, the title could be simplified to more precisely convey the key techniques, cell type, and findings. For example: 

"Single-cell transcriptomic analysis reveals the molecular profile of Go-Opsin photoreceptor cells in sea urchin larvae."

We agree with the reviewer and changed the title accordingly.

Moreover, suggest that the methods and peripheral results that could potentially go in supplementary info include:

- The detailed statistical tests for the scRNA-seq analysis (keep summary in main text).

We agree with the reviewer, therefore we moved the detailed description of the statistical tests for the scRNA-seq analysis in the Supplementary S2 and kept a summarized version in the main manuscript. 

- The circadian rhythm gene analysis, since the link to Opsin3.2 cell function is unclear.

Although the evidence of a link/involvement of the Opsin3.2 positive neurons to the circadian rhythms is unclear, we believe that this was an important point to investigate and we prefer to keep this analysis in the main figure.

Finally, some long complex sentences, like this example in the Discussion : "In any case, intriguing is the expression, although in a small percentage of cells and at low levels, of many genes annotated as phosphodiesterases (PDEs), which might be indicative of a hyperpolarizing cascade, similarly to what happens in mammalian eye photoreceptor cells..." could be divided into shorter sentences to enhance readability. 

We agree with the reviewer and attempted to break long sentences into shorter ones.

In summary, this paper solidly demonstrates the molecular signature of an understudied photoreceptor cell type, but the research represents an incremental advance in a specialized area rather than a major leap forward. With minor revisions to highlight the most salient results, it will make a nice addition to the literature on photoreceptor diversity.

Thank you